# Exploring barriers of household contact screening of index case contacts of pulmonary tuberculosis cases in Sekela district, Amhara region, Ethiopia: 2023; descriptive qualitative study

**Mihrete Geremew**[1]*, **Zewudu Dagnaw**[2], **Eskezyiaw Agedew**[1], **Abiot Aschale**[1], **Aysheshim Asnake Abneh**[1], **Tadele Derbew Kassie**[1]

1 Public Health Department, College of Health Sciences, Debre Markos University, Debre Markos, Ethiopia, 2 Environmental Health Department, College of Health Sciences, Debre Markos University, Debre Markos, Ethiopia

* mihretgeremew60@gmail.com

## Abstract

### Introduction

Tuberculosis (TB) remains a leading global public health challenge, with about one-third of the world's population infected and at risk of developing active disease during their lifetime.. Contact screening is crucial strategy for active case detection and to identify more cases. It involves systematic screening of the contacts of known TB patients. There is an ongoing need for research into barriers to contact investigation to better information uptake.

### Objective

Exploring the barriers to Household Contact screening of pulmonary Tuberculosis Cases.

### Materials and methods

A descriptive qualitative study was conducted at Sekela district, West Gojjam zone, Ethiopia. Purposive sampling (heterogenic) technique was used to recruit study participants. Fourteen participants were involved in the study in accordance with data saturation which includes health extension workers, PTB patients; household contacts of TB patients, health center TB focal and district TB officer. Data was collected through in-depth interviews using a semi-structured guide, transcribed word by word and conceptually translated. A thematic analysis was conducted after coding to answer specific study questions.

**Data availability statement:** All relevant data are within the manuscript and its Supporting Information files.

**Funding:** The author(s) received no specific funding for this work.

**Competing interests:** I have read the journal's policy, and the authors of this manuscript have the following competing interests: The authors have declared that no competing interests exist.

## Result

The main barriers for contact screening of PTB case explored were; health care system related barriers like not conducting review meeting power, lack of training, lack of supervision and follow up; barriers from the health care workers; work over load, non-commitment; socio economic barrier, discrimination, preference of traditional healers and culture, difficult geographic area; and barriers from patients and contacts are lack of awareness, low health seeking behavior.

## Conclusion

The overall explorations of this study identified multiple and interconnected barriers that range from individuals to health system levels which influence contact screening of pulmonary tuberculosis cases in the study area. Not paying attention to contact tracing activity by health system care system results lack of commitment of health care workers not giving health education to community.

## 1 Introduction

### 1.1 Background

Tuberculosis (TB) continues to be a major public health problem worldwide, ranking among the top causes of death along with HIV. Each day, thousands of people lose their lives to TB, despite the availability of effective preventive and curative measures [1].

Active identification of TB cases is essential for reducing transmission and improving treatment outcomes. It involves systematic screening of the contacts of known TB patients and enables early detection of TB cases and prompts initiation of treatment thus reducing the disease burden, the risk of TB transmission and poor treatment outcomes. Currently, there are millions around the world became disabled and dead due to TB as a result of lack of concentration of Nations for prevention and control of tuberculosis (TB) [2–4]. Presentation of patients by themselves to health facility as a method of control of TB is the indication of the World Health Organization's (WHO) target of eliminating TB by 2035, defined as reducing annual TB incidence to below 10 per 100,000 cannot be achieved [5]. This way of case finding leads highly spread of case in which TB patients stay within the community asymptomatically, but infect others continuously [6–8]. That is why the strategy of contact screening actively in base of community is set as means of case findings [9]. The definitive way of prevention of transmission and reducing death and disability due to TB,is addressing home to home of the community by the health workers to search and find cases and suspects of tuberculosis [10] as forwarded by WHO in 2012 [11]. Although active searching of contacts of index cases is best practice recommended by world health organization, only some countries have practiced such ways of intervention [12], in contrary to developed implementation guidelines of household contact investigation [13]. Due to this reason there were 10 million individuals around the globe infected

by TB in 2017. for better success of reducing morbidity and mortality by TB" END TB strategy" was developed globally and Ethiopia has adopted it [14]. This strategy relies mainly on identifying unreached 90% TB cases at the community and household level [15] in accordance with active case finding [16]. Global end TB strategy adopted by Ethiopia is done mainly by health extension workers at house hold level with support of community leaders for early identification and referral of presumptive TB [17], though the strategy implementation is low still now at national level in 2017 [18].

For any type of TB index case transmission is higher in the HH contacts other than outer communities mainly in the first 2 years of exposure [19] for the reason that most patients with TB are asymptomatic but infect [20]. Evidence indicates that active community level screening of contacts of households is practical and can improve TB cases finding [21–24] yet it remains underutilized in many high burden countries [25] that leads to low contact tracing of families of index cases for finding active TB cases or latent TB infection facilitates to the rapid transmission of TB within the large populations [26,27]. This negligence and low utilization of systemic active contact screening method exposes to high transmission and prevalent of TB including our country Ethiopia which was 7.8% prevalent rate of PTB among contacts [28].To combat this problem, in middle and high TB-burden settings, where Ethiopia is part of it, active case finding and contact tracing of index TB cases is recommended for household and close contacts of infectious TB cases [27,29–31]. It is the very crucial intervention to detect TB cases earlier for treatment and contacts for prophylaxis care before developing active TB [26,32].There is limited evidence about this practice in Ethiopia specifically in our study area except little interventional studies conducted in southern Ethiopia which it also shows low practice of contact tracing [21,24,33].

Enhancing the effectiveness of contact tracing necessitates an in-depth understanding of the factors that hinder its implementation. Only a few studies have examined the elements that can find barriers to Household contact tracing of index tuberculosis cases in Ethiopia and no studies was done in the study area in which TB from contact history is prevalent. Therefore, this study is aimed an exploration of potential barriers to contact tracing of index TB cases in Sekela district, North West Ethiopia.

## 2  Objectives

• To explore barriers of House hold contact screening of pulmonary tuberculosis cases in Sekela district 2024.

## 3.  Methods and materials

### 3.1  Study area and period

This study was conducted in Sekela District, North West Ethiopia from February 1toApril 30,2024. Sekela district is one of the 18 districts found in West Gojjam Zone which is 443 km far from the capital city, Addis Ababa and 160 km from Bahir Dar. According to 2022 population projection the district has total population of 189464 of which 92,837 (49%) are males and 96,627 (51%) are females. Within the District, there are 8 health centers, 35 health posts, which provide treatment for fifteen PTB cases.

### 3.2  Study design

This was a descriptive qualitative study which was carried out from February 1 to February 30, 2023.

### 3.3  study participants and sampling procedures

For this study, study participants were selected purposely (A heterogeneous purposive sampling technique was used) and, contacts and index case with age less than eighteen and seriously ill participants who were unable to communicate and unable to give full information at the time of data collection was not selected. Study participants were from urban and rural; male and female; those who can read and write and who do not read and write. To ensure social

representativeness, a range of participants were involved, including the TB Woreda-level coordinator, focal persons, contact cases, and index cases. These participants were recruited from five health center (which had PTB cases in treatment currently) catchments considering their ability to generate detail information on TB contact screening. This study recruited TB focals and health extension workers from health institutions (health centers and health posts) which were providing DOT during study period/ in health posts and health centers that had PTB case based on their work experience. Pulmonary TB patients were selected from both intensive and continuous phases, and from household contacts that were not screened. Household contacts were recruited that were not screened for tuberculosis, and district TB officer was recruited due to responsibility on the TB program who was able to provide detailed information about contact screening.

Using heterogeneous purposive sampling, fourteen participants who include three TB clinics focal, one district TB officer, four health extension workers (HEWs), three index TB patients and three household contacts of active TB patients were interviewed. The maximum size of participants was determined based on the saturation of responses. Data was considered saturated when the incoming data adequately answers the question, data are complete and no new information comes from interviewing different participants. Redundancy (data saturation) was considered when three people overlap in essence of their responses. This threshold was established based on consensus of teams guided by prior published literatures [34].

### 3.4 Qualitative data collection tools

In-depth Interview Guide; semi-structured guide was used. It was prepared in English and then translated to local languages (Amharic) and back-translated into English by another independent translator. The tool was designed to guide the content of interview of the following issues; explore the multiple layers of barriers that influence contact tracing of index cases. It was prepared to explore patient and household as well as TB focal persons and health extension workers experience on the barriers of contact screening.

Key informant interview guideline: it was developed and used for data collection from the program officer.

### 3.5 Data collection methods

Both In-depth interviews (IDI) and key informant interviews were used. Documents were also reviewed and used to collect data on the status of contact tracing of index TB cases from district health information system (DHIS) of five months and registration books; which was carried out before any data were collected to understand the local context in the status of contact investigation.

The interviews were conducted at the participant's preference setting after discussing their comfortable time and place. TB focal persons and health extension workers were interviewed at their working place. The index TB patient and three household contacts were interviewed at their homes. After explaining the purpose, risks, benefits of the study, and the length of the interview, the participants were asked for their consent in order to participate in the study. They were committed and voluntary to be interviewed with a smile face of accepting the interviewer. Then the interviewer asked permission to start the interview and audio-tape recording to ensure verbatim transcription. The interviews ranged from 23–42 minutes. Notes were taken during each interview and used as backup records in case of accidental data loss or damage; however, no data loss or damage occurred during this study. All interviews were conducted by the investigator in which one in-depth interview was performed per day. Key informant interview was held after in-depth interview was finished to triangulate data. The interviewer used probing technique, by using how and why, to get adequate data on the point of interest.

### 3.6 Data analysis

Data was analyzed iteratively. It mainly focused on collected data in the transcripts of recorded interviews and from field notes. The digitally recorded data was listened carefully and transcribed into local language (Amharic) and contextual

translation was made. Nonverbal cues, including facial expressions, tone, and signs of hesitation or frustration, were noted to contextualize responses and guide interpretation, though not all are reported verbatim in the Results. Transcripts were read three times to gain a comprehensive understanding of participants' meanings and to identify emerging themes and sub-themes. After we read, primary coding was done. Primary coding was done line-by-line review of transcripts, followed by axial coding to identify key words and phrases. Data were displayed to examine variations across themes, and then reduced to capture the overall dataset. Narrow themes were merged into broader ones, and essential themes were distinguished from non-essential. Thematic analysis was used throughout, with the principal investigator interpreting reduced themes to reveal core meanings of experiences. Participant quotes were included to illustrate categories and their connections to themes. Open Code software (version 4.02) facilitated the analysis.

### 3.7 Trustworthiness of the study

Data was collected from diversified study participants recruited from different settings that have relevant experience. To assure social representativeness a range of participants were involved, TB Woreda level-coordinator, focal, contact and index cases. The principal investigator was spent long periods in the fields to build trust with the study participants throughout the process of the study. Member check (Sharing findings with participants to confirm accuracy and ensure their views are reflected) was done to assure the credibility of this study. To assure confirmability (ensuring results reflect participants' perspectives, not researcher bias, through careful documentation), an audit trail was employed by emphasizing thick descriptions of the study process from the inception to the final report. The recorded interviews were not deleted to enable others track the process. To ensure dependability, data analysis, interpretations, and conclusions were continuously peer-reviewed by three colleagues.

Transferability was ascertained by providing evidence based, detailed explanation of the study starting from sampling to data analysis. The data obtained was kept confidentially with code of participants rather by their name and informed consent was taken before starting of the in-depth interview with each participant.

### 3.8 Ethical considerations

Ethical approval was obtained from Debre Markos University, Ethical Review Committee of college of medicine and health sciences with IRC protocol no. HCS/R/C/ser/PG/Co/03/11/15/ on date 30/12/2022 and permission obtained from the District Health Office. Written consent was requested from each study participant and this written consent kept in recorded for witness prior to the commencement of data collection. Helsinki declaration was followed.

## 4 Results

### 4.1 Socio-demographic characteristics

Nine men and four women were participated in the in-depth interview and one woman in the key informant interview. The age of the participants ranged from 23 years old to 40 years old. All HEWs from IDI and one TB patient were College diploma and above whereas three TB focal and one contact had first degree and, TB officer has Masters but two contacts and 2 patients can read and write. Seven of the participants were civil servants and urban dwellers others were from rural. All of the study participants were Orthodox Christian (Table 1).

### 4.2 Barriers to household contact tracing of index tuberculosis cases

In this study, four themes with sub-themes were explored. These four themes are; Health care systems, socio-cultural and economic, Health care providers' and index case and their contacts. The data analysis was adapted from the thematic analysis to interpret the data and find meaning in the participant's experience and identify barriers that could be improved for TB contacts screening (Table 2).

**Table 1. Socio-demographic characteristics of the participant in Sekela district, Amhara, Ethiopia, 2024.**

| Code | Sex | Age | Educational status | Marital status | Place of residence | Occupation | Phase of treatment of patients and no. of contacts respectively |
|---|---|---|---|---|---|---|---|
| Patient1(P1) | M | 23 | Didn't attend | Single | Rural | Farmer | Continuous, 4 |
| Patient2 (P2) | M | 36 | Diploma | Married | Rural | Civil servant | Intensive, 3 |
| Patient3 (P3) | M | 27 | 10th | Married | Rural | Farmer | Continuous, 5 |
| Health extension worker 1(E1) | F | 33 | Diploma | Married | Rural | Civil servant | |
| Health extension worker (E2) | F | 38 | Diploma | Married | Rural | Civil servant | |
| Health extension worker (E3) | F | 34 | Diploma | Married | Rural | Civil servant | |
| Health extension worker (E4) | F | 35 | Diploma | Married | Rural | Civil servant | |
| Officer (O) | F | 40 | Master | Divorced | Urban | Civil servant | |
| Contact (C1) | M | 28 | Didn't attend | Single | Rural | unemployment | |
| Contact 2 (C2) | M | 27 | Grade 6 | Single | Urban | Merchant | |
| Contact 3 (C3) | M | 25 | Degree | Single | Rural | Un employed | |
| Focal person (F1) | M | 32 | Degree | Married | Urban | Civil servant | |
| Focal person (F2) | M | 30 | Degree | Married | Rural | Civil servant | |
| Focal person (F3) | M | 31 | Degree | Married | Rural | Civil servant | |

M = male, F = female.

### 4.3 Theme 1: Health system related barriers

These are barriers originated from the health care governing body that are not fulfilling the expected supply, man power and budget. Contact screening activity is led by the government. If the necessary conditions are not fulfilled by the government, performing contact tracing becomes very difficult. This study explored a set of governmental factors that undermines TB contacts screening. There are nine sub-themes under this theme. These are;

**4.3.1 Not conducting review meeting.** Conducting regular and periodic review meeting with stakeholders is very crucial activity which motivates and alarms, focal and health extension workers for better tracing. Majority of the key informant interview stated this.

*"There is no review meeting even once a year. Evaluation is in general, it means that at the district level, we will not review the TB level itself quarterly". A 30 year male TB focal*

**4.3.2 Lack of manpower.** Fulfilling trained manpower is the essential condition in order to serve the community quickly and to conduct tracing of contacts effectively. In this study area, health institutions have not enough manpower that performs all activities of the health post by a single health extension worker and all activities of the health center by single laboratory profession which affects the contact tracing activity.

Almost all participants explained that there are no enough health workers in each institution which leads a single man works all-round the health center as TB focal, outpatient department and other departments; This results in neglect of TB contact screening. Working all rounds the health center and health post by single laboratory and single health extension worker, and not assigning focal for only TB activity is the main barriers for contact screening as low working power which result over work load of HCW and delay in the TB contact screening.

*"In addition to this there is lack of man power, I am TB focal but I also work in under- 5,outpatient department injection class it is difficult that the current situation of the HC in which the lack of man power is not solved. Next to this in our cluster there is only one lab technician and a single HEW in each health post which is loaded by vaccination and maternal cases which makes them difficult to concern on contact screening." A 31 years old male TB focal F3*

**Table 2. Themes and subthemes and categories of the explored barriers of contact tracing of index cases in Sekela district, Amhara, Ethiopia 2023.**

| Themes | Subthemes | Categories |
|---|---|---|
| Health system related | Health sector related | Not conducting reviewing meetings<br>Lack of manpower<br>Lack of adequate Budget<br>Lack of Training<br>Lack of supply<br>Absence of close Supervision<br>Poor relationship<br>Delay in service/long waiting time<br>Absence of separated class for contact and index TB cases |
| | Health care worker related | Commitment problem<br>Lack of awareness<br>Work over load |
| Community related | Contact and index case related | Lack of awareness<br>Giving Priority for their daily work<br>Lack of commitment |
| | Socio economic and cultural | Distance, up and down Landscape<br>Discrimination<br>Long distance<br>Lack of money for transportation/low economic status<br>Preference of traditional medicine and wrong belief |

**4.3.3 Lack of adequate budget.** Health care workers are expected to screen household contacts by going up to the homes of the index cases. Reaching remote areas to screen for contacts was difficult as participants described as there is no regular transport and adequate budget was not allocated for available motor transport.

*"there is no adequate budget for transportation to monitor and evaluation and support health posts and health centers, for review meeting as well as to build TB class". 40 years female officer*

*"There is no allocated budget for transportation to address the remotest areas which is difficult to reach on foot." A 33 years old health extension worker*

*"There are also transportation and budget constraints to go to their homes to trace contacts and give health education about the TB situation to those who are far away from the community." (A 31 years old male TB focal*)

**4.3.4 Lack of training.** Updated training is critical for health care workers to screen contacts and to find cases actively. Some participants in this study remarked that health workers were not trained for TB management and contact screening others not updated. This has prevented patient's family/contacts from getting TB screening services.

*"...not enough training is being given, That is, when a professional is left, another untrained professional will provide services. And there is a lack of training. Other health extensions are not trained in TB. There are labs in all seven rural areas that have not received TB or TB training" (40 years old female TB officer)*

*"Third, professional training is not provided. Now I am the only one who has taken from the institution and I have taken before. Now the health care workers did not know what the update is." A 31 years old male*

**4.3.5 Lack of supply.** Resources were essential to carry out for screening and diagnosis of tuberculosis. Shortages of reagents, papers, interruption of electricity were the listed supply related barriers. Participants explained that they

were turned repeatedly due to lack of working reagent and they were delayed to bring their family to health institutions for screening.

*"Even though my families have gone to health institutions on not working day,they turned without screening due to no working resource" a 23 years old male patient.*

*"There was the incontinency of supplies like that of reagent, falcons tube microscope were stock out ......" (A 40 years old female TB officer)*

**4.3.6 Absence of close supervision.** Continuous Monitoring and supervision over health center and health post by checklist improves contact screening performance and build ups the awareness of health care workers, which also increase the confidence of contacts to go to health institutions if they are avail on time on work place. This study also identified that no supervision of health works by government was the barriers of contact screening. Participants mentioned that if there were no supervision, absence of health workers especially HEWs from work place makes them bored and results repeated loss of their working time.

*"Continuous and strict follow up and supervision of the HCWs is very essential, I was bored because of coming from distant areas and I cannot get the HEWs and their service."a 23 years old male patient p1*

*"I don't think health professionals are doing much monitoring and control. The health institutions are closed, I see, they don't go out or come in at on time."* (A 36 years old male patient)

**4.3.7 Poor relationship.** In order to perform contact screening effectively, strengthening connection between health institutions and the office and between health centers and health is the main strategy of the health system. But In this study, Participants explained that the linkage among woreda health office, health center and health post was poor/ inconsistent which result continuous feedback and super vision was not given.

*"Not giving feedback, by saying this is a weakness, this is a strong point; Not being identified as". (*A 38 health extension worker*)*

*"Due to lack of paper and band land scape of woreda as well distance from the center there is no enough feedback given to health centers." 40 years old TB officer*

*"Not giving feedback on failed tasks, not giving feedback on the existing things that have been done or not". (A35 years old HEW)*

**4.3.8 Delay in service/ long waiting time.** After reaching the health institutions with facing challenges there is delay and long waiting time for getting service which bores the contacts and patients. Quickly serving of index cases and contacts helps the reduction of transmission of TB to others. Here, participants said that in health institutions waiting for a long time was their barriers especially for contacts and index cases.

*"Getting service after going to the health center was difficult; we wait for a long time to there." (a 27 years old male patient)*

*"Ayiii…[local Amharic language] not even they come to our home for screening, they do not care on the health institutions after we go." (A 36 years old male TB patient P2)*

*"Apart from that, when we send them away quickly, they go there quickly and are not treated at the time of reaching. If they are not treated quickly, the problem of abuse will appear." (A 38 years old HEW)*

**4.3.9 Absence of separated class for contacts and index TB cases.** Index cases and suspects shall be served in separated class from the general populations to prevent transmissions. But in our case, there is lack of clean and separated and standardized class for coughers and TB patients in most HCs of the woreda. The roof, floor and wall of the available TB class are cracked broken and non-ventilated. This is another difficult issue for coughers and contacts to screen which facilitate transmission of TB.

*"If you see TB room, you will be surprised. It's just something in the field and not attractive. I am doubtful with the drug I used by it because bird may enter the room and discard." (A 36 years old male patient)*

*"There is no separated waiting class for contacts. The TB room is the service room; even though the class has been separated from the examination room, it has not been given enough classes. It's very narrow with no window, so it's not enough." (A 40 years old female TB officer)*

**4.4 Theme 2: Health workers related barriers**

These barriers are originated from health care providers occurred as result of not paying attention by government. There are three sub themes under it:

**4.4.1 Lack of commitment.** This study identified that the health workers consider TB as second job and works other classes as constant and TB clinic sometimes only as well as some health workers push only to focals. This leads to ignorance of contact screening. Considering the activity of contact screening as their own job must be practiced by all HCW to be effective in the control of TB and active case finding activity.

*"Other HCW who works under five, maternity, emergency assume that the work of TB is for only focal and they do not care about contacts and cases." (A 33 years old TB focal)*

*"There is a lack of commitment with the professional. Lack of concentration: work as a second job."(A 40 years old female TB officer)*

**4.4.2 Lack of awareness.** Training must be given for non-trained health workers and updating is needed for old trained in order to capable for screening of contacts of index cases. But some participants express that they were not trained for TB and don't have known how contact screening is done.

*"Health extension workers were not trained about TB management and screening of contacts, so they have not aware-ness to them."(A 32 years old TB focal)*

*"We have no aware, Because of that, the government is cold and do not pay attention to it, so we don't get much train-ing." (A 35 years old health extension worker)*

**4.4.3 Work overload.** In order to facilitate effective TB contact screening and to control the disease as a whole assigning focal to the work of TB only is very important. But most institutions assign focal for more than 2 case teams and then contact screening is difficult due to work over load.

In addition to this there were only one laboratory professionals who work the whole activities of the health center with no time to do screening of contacts.

*"There is no delegated person who cares to me and gives anti TB drugs in the TB class; he comes after calling from adult outpatient department". (A 36 years male patient P2)*

*"Another, as I explained before from lack of laboratory professionals and ours is not enough health extension workers. I am the only one who works in this health post. Currently, there are more than 18 tasks of extension. There is work over load to maintain the quality and to carry them out. Workload is one of the challenges." (A 35 years old health extension worker)*

### 4.5 Theme 3: Patient/index case and contact related barriers

Barriers from the index cases and contacts that includes three sub themes.

**4.5.1 Lack of awareness.** Creating awareness and health educations are the main tools for prevention and control of tuberculosis which is the essential activities of HEW. HCWs are expected to give information to patients that should be taken as a measure to prevent TB. But these participants identified that they didn't know whether the families are screened or not which lowers the contact screening progress due to not awareness were not created. Contacts and the index cases assumed the only treating of the index cases.

*"I do not know whether my families are screened or not." a 23 male patient P1*

*"The other thing from the perspective of the patients and their families is that there is a problem of awareness. There is a lack of awareness. We did not make them aware of it" a 33 years old HEW*

*"….now regarding TB patients and their families; we have no awareness; from our point of view. Now we hardly have the awareness to go and be screened for being contagious. It means that we don't have much awareness." A 25 years old male contact*

**4.5.2 Giving priority for their daily work.** As there is no difficulty to work due to disease, farmers give priority to their work over screening of contacts and they don't consider TB was a problem to their family.

*"The patient and their family were not voluntary to be screened because they give priority to work and attention was not given about screening of TB, even if they were made aware of it, the problem is that they are reluctant to do it." a 33 years old health extension work*

*"Work over load of family; giving Priority for work." A 25 years old male contact*

*"Give priority to their work. It means they don't want to go health institutions and be screened, give priority to their work.." (A 38 years old health extension worker)*

**4.5.3 Lack of commitment/losing hope.** When the patient and contacts assumed their cough or their case is TB and there is negative of microscope patients lose hope. They always think there is TB when they cough out which may be other respiratory case. They are also reluctant to be screened even though they have awareness because they are symptom free currently.

*"patients and contacts lose hope when there is symptom positive and microscope is negative and says you do not get my case so I do not want to come to health institutions." A 33 years old male focal F1*

*"Mainly we have no motivation. We have no commitment". A 25 years old male contact*

### 4.6 Theme 4: Social, cultural and economic related barriers to contact screening

For effective contact screening the contribution of the community is the main issue. Supporting the index cases and initiation for screening is expected from the community. In the study setting there are four explored barriers under this.

**4.6.1 Distance and up and down landscape.** Most kebeles of the district are wide and far from health institutions which makes them difficult to address the contact screening service. There was also the remotest area which includes Mountains Rivers, gorges, abyssal, up and down hills. This is the problem to go and screen the contact or difficult for contacts to come and screened.

*"patients who are in DOT are scarified due to long distant from institutions and asks to take medications of intensive phase for more than three days" a 33 years old male TB focal*

*"The distance is too far, the road is too far, and for example, it takes me about 3 hours to get to the health center. The road is far from the health center. There is no transportation because the area is far away." (a 23 male patient)*

*"The area where the people are now is one of the most remote places, and those people now find it very difficult to bring their families to get tested for TB. A 33 years old HEW*

**4.6.2 Discrimination.** There were still discriminations to tuberculosis patients by their neighbors and communities that made difficult and afraid them to bring their family to health institutions for screening and preventing the patients to participate in social activities.

*"Oh, since I was told I have TB, there is isolation from the community. There is discrimination on me. My father said me no to go (zikr mahiber) after the community said to him not to send". A 23 male patient P1*

*"So they are now left alone; to keep the secret; But if every family went to be investigated, the society will isolate us; they think because they are TB patients ….they discriminate us because they think that." A 33 years old HEW*

**4.6.3 Lack of money for transportation/low economic status.** In our setting naturally the District has ups and downs gorges, mountains and rivers which is very difficult to address such type of area. Infrastructures are not nearby communities, to go to health institution for screening it is far from home and the transport was not available, even though motor transportation is there, it was so costly and not affordable for patients and their family.

*"We have no enough money to come 'to health institutions by motor because it is far from our home and no transportation." A 30 years old male contact C2*

*"It is not affordable to me to bring my family to health centers by car or motor." A 25 years old TB patient P3*

**4.6.4 Preference of traditional medicine and wrong belief.** In the community there were assumptions about PTB as cancer and treating it with traditional medicine is better rather early screening and caring in health institutions, which result delay in seeking health service and transmission of TB and death of the index cases.
Participants mentioned that preference of traditional medicine and wrong thought, and going to traditional healers over health institutions was another barrier for contact screening.

*"In the culture there are individuals who are valuable in the society and magicians, sometimes TB patients and their contacts prefer to go there."40years old 17 years' experience TB officer*

*"There are many people who are investigating the traditional medicine area, saying that it is TB, it is gout. There are many people who have to go there". A 36 years male patient P2*

## 5 Discussion

The 2015 global End TB Strategy calls for bold, new, patient-centered, active case-finding strategies. Exploring the barriers of contact screening of TB index cases will be critical for any of these new approaches to succeed. In pursuit of these goals, we purposely approached front-line owners of this activity in face to face in-depth interview about barriers to household contact investigation in the study area. This study explored health workers and TB program officers as well TB patients and their contacts views on barriers of contact screening facing in the study district. Participants identified barriers related to the health care system, health workers, PTB cases and their household contacts, as well as socio-economic and cultural factors. Social and cultural barriers—such as discrimination—were explored to hinder the early screening of TB contacts. This discrimination behavior of the community results fear of patients to take drugs and send their all contacts to health institutions for screening and sometimes they close themselves. The study explored that contacts and patients experienced discrimination from their neighbors, consistent with studies conducted in Uganda, which also identified discrimination and stigma as major barriers to uptake clinical evaluation at a health facility [35,36]. In contrast to this other studies conducted in south Ethiopia did not showed that discrimination as a barrier [37]. The difference might be occurred due to difference of socio-cultural differences. Preference of traditional medicine and wrong thought like that of *yebet himem* was another cultural factor which was an obstacle of contact screening that was found in this study. This finding was similar with findings of studies conducted in Madang province Guinea which found that Participants claimed that on many occasions patients refused to take the treatment and went home to seek treatment from traditional healers in their villages [38].

The other socio economic and cultural barriers this study found was the distance of contacts home to reach the screening center that made them to give up of traveling for long and affects the contact tracing activity. This finding aligns with studies from Vietnam and Brazil, which reported that the long distance between contacts' homes and screening facilities hindered contact screening [39,40]. Lack of money for transportation was identified as a key socio-economic barrier that hindered contact tracing. This finding was similar to other studies which found that limited financial resource is a common obstacle to be investigated and traced for TB [37,41,42].

Prolonged waiting times at health facilities pose a significant barrier to the screening and investigation of household contacts of TB patients. This observation aligns with findings from a study conducted in Uganda, which similarly identified long waiting times as an impediment to contact tracing, attributable to health worker overload and limited commitment [35]. Unavailability of materials and chemicals such as reagents and interruption of power were also explored to be barriers for contact tracing of index TB cases. Shortage of reagents results repeated appointment and waiting for a long time which leads give up of contacts. This result is supported by reports conducted in Botswana and Oyo State, South West Nigeria where poor or inadequate facilities, poor medical infrastructure, and limited laboratory identification were stated as obstacle to contact tracing screening services in resource-constrained settings [42,43]. This report revealed that lack of human power at health facilities as a barrier that affected TB contact tracing. This report is in congruent with a study conducted in Canada [36], Thailand and Myanmar [44] as there was often no permanent or full-time nurse or shortage of human power available in a given community to implement TB programming.

A shortage of trained personnel could leads to work overload among health workers and inadequate contact tracing, resulting in missed opportunities for early screening and delayed identification of secondary active TB cases, thereby hindering timely detection and treatment of infected contacts [30].

Absence of supervision and monitoring to health centers and health posts were explored as a barrier to contact tracing of index TB patients. This study finding agrees with a study conducted in South Africa that showed a lack of adequate supervision and direction from the district or provincial TB managers [45]. The World Health Organization (WHO) recommended doing continuous monitoring using standardized procedure [46]. Monitoring and supportive supervision can contribute to health worker's awareness, by filling the gaps which occurs due to lack of awareness on time so that they effectively perform their duties [31].Therefore, regular and persistent monitoring and supervision of TB and contact tracing

activity alone by concerned body with a detailed checklist is the main method of enabling of health workers for contact tracing of index TB cases.

This study explored from the contacts and index cases related barriers revealed that lack of information about screening of contacts. The same finding was identified in study conducted in Uganda Kampala which explored that TB patients did not know about whether contacts traced or not [35]. This resulted that index case would transmit to his contacts and showed that awareness creation by health workers was not conducted to contacts and index cases. Another barrier related to a patient in the current study for not screened was giving priority for work due to workload over going to health facility for tracing. In accordance with this finding, studies conducted in Uganda also showed that workload was one of the barriers identified for TB contacts' not being screened by going to the health institutions [35].

The barriers related to health workers were the lack of commitment due to not giving training and work overload to contact tracing that compromises the screening activity. This finding was in line with the finding of the study conducted in Botswana which showed that motivation and commitments were barriers for implementation of contact tracing [43].

## 6 Strengths and limitations

The strength of this study was the data source which was primary data collected directly from the study participants by IDI without any idea restriction of the respondents that makes it more accurate and helps to explore the most obstacles that prevent screening of contacts of pulmonary TB cases. As a limitation of this study was didn't triangulated by method and not used different method of data collection due to lack of time.

## 7 Conclusion and recommendation

The overall explorations of this study identified that many barriers undermine the tracing of contacts of tuberculosis index cases in the study area. These multiple and interconnected barriers range from the individual to community and the health care system showed that the activity of TB specifically contact screening was not given attention by health sector governing body. Following to this the owner of contact tracing the health care workers also were not committed on providing health education about the necessity of contact screening to the community, that results lack of awareness of contacts and discrimination by the community. Therefore, the local government needs to fulfill different resources and continuous monitoring and evaluation on screening and monitoring contact case screening.

## Supporting information

**S1 File. Translated and transcribed data.**
(ZIP)

## Acknowledgments

We would like to thank Debre Markos University College of Health Science Research and Community Service for giving us to conduct this research. We also acknowledged Sekela district health office, and study participants for giving us important data.

## Author contributions

**Conceptualization:** Mihrete Geremew, Tadele Derbew Kassie.

**Data curation:** Mihrete Geremew.

**Formal analysis:** Mihrete Geremew, Tadele Derbew Kassie.

**Funding acquisition:** Mihrete Geremew.

**Investigation:** Mihrete Geremew, Aysheshim Asnake Abneh.

**Methodology:** Mihrete Geremew.

**Project administration:** Mihrete Geremew.

**Resources:** Mihrete Geremew.

**Software:** Mihrete Geremew.

**Supervision:** Zewudu Dagnaw, Eskezyiaw Agedew, Abiot Aschale, Tadele Derbew Kassie.

**Validation:** Mihrete Geremew, Eskezyiaw Agedew, Tadele Derbew Kassie.

**Visualization:** Mihrete Geremew, Abiot Aschale, Aysheshim Asnake Abneh, Tadele Derbew Kassie.

**Writing – original draft:** Mihrete Geremew.

**Writing – review & editing:** Mihrete Geremew, Aysheshim Asnake Abneh.

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
