## [Decision Letter · Decision Letter 0]

29 Oct 2025

Dear Dr. geremew,

We look forward to receiving your revised manuscript.

Kind regards,

Frederick Quinn

Academic Editor

PLOS ONE

https://journals.plos.org/plosone/article?id=10.1371%2Fjournal.pone.0233358

In your revision ensure you cite all your sources (including your own works), and quote or rephrase any duplicated text outside the methods section. Further consideration is dependent on these concerns being addressed.

Additional Editor Comments (if provided):

Reviewers' comments:

Reviewer's Responses to Questions

**Comments to the Author**

1. Is the manuscript technically sound, and do the data support the conclusions?

Reviewer #1: Yes

2. Has the statistical analysis been performed appropriately and rigorously?

Reviewer #1: N/A

3. Have the authors made all data underlying the findings in their manuscript fully available?

Reviewer #1: Yes

4. Is the manuscript presented in an intelligible fashion and written in standard English?

Reviewer #1: No

Reviewer #1: This manuscript addresses an important topic related to tuberculosis (TB) contact investigation and barriers to effective implementation. The study provides valuable qualitative insights from multiple perspectives, including healthcare system, health workers, and patients. Overall, the structure of the Methods and Results sections is appropriate, and the thematic analysis approach is suitable for the research objectives.

However, several areas require clarification, consistency, and refinement to improve readability, methodological transparency, and the rigor of reporting. The following specific comments are provided to guide revision.

1. Section 3.1 – Study Area and Period

• Add a space in “96627(51%)” → should read “96,627 (51%).”

2. Section 3.3 – Study Participants and Sampling Procedure

• Define the abbreviation HEWs at first mention.

• Clarify why redundancy was defined by responses from three people (“Redundancy was considered to occur when three people overlap in essence of their responses.”). Explain whether this threshold was based on prior qualitative literature or a team consensus.

• Suggested sentence revision for clarity and grammar:

“To ensure social representativeness, a range of participants were involved, including the TB Woreda-level coordinator, focal persons, contact cases, and index cases.”

3. Section 3.5 – Data Collection Methods

• Define DHIS at first use.

• Ensure consistent line spacing throughout this section.

• Add space in “23- 42minutes” → should read “23–42 minutes.”

• The sentence “Notes were taken at each interview that used as backup file if there were lost or damaged records happened accidentally” needs clarification. Please indicate whether any data loss or damage occurred in this study.

• The sentence “It mainly focused on collected data in the ??? and from field notes” appears incomplete — please insert the missing text.

• Clarify how observations of facial expressions, reluctance, emphasis, and frustration were incorporated into data interpretation, as these aspects are not reflected in the Results section.

• Specify how many times each transcript was read during thematic analysis (“Transcripts were read and reread…”).

4. Section 3.8 – Trustworthiness and Quality Assurance

• Correct typographical errors:

o “audit trial” → audit trail

o “confirm ability” → confirmability

• Briefly define member checking and confirmability for readers unfamiliar with these qualitative terms.

• Specify how many individuals participated in the peer-review process for data analysis and interpretation (“The data analysis, interpretations, and conclusions were continuously peer reviewed…”).

5. Participant Characteristics (Table 1)

• Clarify the educational levels described (avoid abbreviations such as “level 4”). Write them out (e.g., “secondary education,” “college diploma,” etc.).

• Correct “where as” → whereas.

• Verify missing information for participant C1 (education, marital status, residence, occupation).

• Clarify the meaning of “Phase of RX of Ps”—what do “RX” and “Ps” represent?

• Remove the religion column if all participants are Christian.

• Consider combining the last two columns into one “Notes” column for simplicity.

• Avoid abbreviations for participant codes.

6. Thematic Findings (Table 2 and Results Section)

• Explain the meaning of “not reviewing” under the subtheme “Health sector–related barriers.”

• Please check the consistency between table 2 and results for each category. You should verify that

o Every subtheme and illustrative quote in Table 2 appears and is discussed in the Results section.

o The wording of categories and subthemes matches exactly (e.g., capitalization, phrasing).

o The order of presentation in Table 2 aligns with how themes are described in the text.

• Correct page numbering starting from the page containing Table 2.

• Revise sentence for clarity:

“If the necessary conditions are not fulfilled by the government, performing contact tracing becomes very difficult.”

• Correct minor typographical issues:

o “This results as ignorance of Tb contact screening.” → “This results in neglect of TB contact screening.”

o “house hold” → household

o “man power” → manpower

o “Up dated” → updated

• Format improvements:

o Bold subtheme heading: “Lack of commitment” under Theme 2.

o Bold full theme headings:

Theme 3: Patient / Index Case and Contact–Related Barriers

Theme 4: Social, Cultural, and Economic–Related Barriers to Contact Screening

**Do you want your identity to be public for this peer review?** For information about this choice, including consent withdrawal, please see our Privacy Policy

Reviewer #1: No

---

## [Author Response · Author response to Decision Letter 1]

10 Nov 2025

This is the point by point response to the reviewers

Thank you dear reviewers, we have made revision based on your valuable comment. We learn a lot from your comment and if additional comment, we are delightful to revise again.

Here are the requested revisions

Response: we revised to comply with PLOS ONE’s formatting and file-naming requirements, following the provided style templates.

Response: we have revised and paraphrased the manuscript

Response: we have made correction on the system

Response: we have made based on the recommendation

Response: we have removed from the declaration section

Response: we have addressed based on their recommendations

Review Comments

Reviewer #1: This manuscript addresses an important topic related to tuberculosis (TB) contact investigation and barriers to effective implementation. The study provides valuable qualitative insights from multiple perspectives, including healthcare system, health workers, and patients. Overall, the structure of the Methods and Results sections is appropriate, and the thematic analysis approach is suitable for the research objectives.

However, several areas require clarification, consistency, and refinement to improve readability, methodological transparency, and the rigor of reporting. The following specific comments are provided to guide revision.

1. Section 3.1Study Area and Period

• Add a space in “96627(51%)” → should read “96,627 (51%).”

Response: we appreciate this comment and we made correction.

2. Section 3.3 –Study Participants and Sampling Procedure

• Define the abbreviation HEWs at first mention.

Response: thank you for this, we have first define it as health extension worker

• Clarify why redundancy was defined by responses from three people (“Redundancy was considered to occur when three people overlap in essence of their responses.”). Explain whether this threshold was based on prior qualitative literature or a team consensus.

• Suggested sentence revision for clarity and grammar:

“To ensure social representativeness, a range of participants were involved, including the TB Woreda-level coordinator, focal persons, contact cases, and index cases.” Clarify why redundancy was defined by responses from three people (“Redundancy was considered to occur when three people overlap in essence of their responses.”). Explain whether this threshold was based on prior qualitative literature or a team consensus.

Response: we corrected the grammar and accepted the given comment. For “Redundancy was considered to occur when three people overlap in essence of their responses”. Explain whether this threshold was based on prior qualitative literature or a team consensus.” We decide this as team consensus based on previous literature

3. Section 3.5 – Data Collection Methods

• Define DHIS at first use.

• Ensure consistent line spacing throughout this section.

• Add space in “23- 42minutes” → should read “23–42 minutes.”

Response: we accept the comment

• The sentence “Notes were taken at each interview that used as backup file if there were lost or damaged records happened accidentally” needs clarification. Please indicate whether any data loss or damage occurred in this study.

Response: there was no lost file in this study, we detail explained in the manuscript as Notes were taken during each interview and used as backup records in case of accidental data loss or damage; however, no data loss or damage occurred during this study.

• The sentence “It mainly focused on collected data in the ??? and from field notes” appear incomplete — please insert the missing text.

Response: we inserted the missed word in the document

Clarify how observations of facial expressions, reluctance, emphasis, and frustration were incorporated into data interpretation, as these aspects are not reflected in the Results section.

Response: we appreciated this helpful suggestion and we have made correction as. Nonverbal cues, including facial expressions, tone, and signs of hesitation or frustration, were noted to contextualize responses and guide interpretation, though not all are reported verbatim in the Results.

• Specify how many times each transcript was read during thematic analysis (“Transcripts were read and reread…”).

Response: we have made revision based on this concept

4. Section 3.8 – Trustworthiness and Quality Assurance

• Correct typographical errors:

o “audit trial” → audit trail

o “confirm ability” → confirmability

c we have accepted this suggestion and made revision, thank you very much!!

• Briefly define member checking and confirmability for readers unfamiliar with these qualitative terms.

Response: We briefly explained this on the manuscript in bracket

• Specify how many individuals participated in the peer-review process for data analysis and interpretation (“The data analysis, interpretations, and conclusions were continuously peer reviewed…”).

Response: thank you we add three colleagues were participated in peer review

5. Participant Characteristics (Table 1)

• Clarify the educational levels described (avoid abbreviations such as “level 4”). Write them out (e.g., “secondary education,” “college diploma,” etc.).

Response: thank you very much we have made correction

• Correct “where as” → whereas.

• Verify missing information for participant C1 (education, marital status, residence, occupation).

Response: yes we missed and we add the missed

• Clarify the meaning of “Phase of RX of Ps”—what do “RX” and “Ps” represent?

Response: Rx=treatment and pts=patients we fully explained in the document

• Remove the religion column if all participants are Christian.

• Consider combining the last two columns into one “Notes” column for simplicity.

Response: Since merging and labeling “Notes” may confuse readers we merged and replaced with phase of treatment and number of contacts respectively

• Avoid abbreviations for participant codes.

Response: thank you we avoided the abbreviations and wrote in full form

6. Thematic Findings (Table 2 and Results Section)

• Explain the meaning of “not reviewing” under the subtheme “Health sector–related barriers.”

Response: thank you it is to mean not reviewing mean not undertaking review meeting

• Please check the consistency between table 2 and results for each category. You should verify that

o Every subtheme and illustrative quote in Table 2 appears and is discussed in the Results section.

Response: thank you very much we made correction; there were ordering problem in the table and result section

o The wording of categories and subthemes matches exactly (e.g., capitalization, phrasing).

Response: we separteded themes and subthemes

o The order of presentation in Table 2 aligns with how themes are described in the text.

• Correct page numbering starting from the page containing Table 2.

Response: We revised based on the comment, thank you

• Revise sentence for clarity:

“If the necessary conditions are not fulfilled by the government, performing contact tracing becomes very difficult.”

Response: we appreciated the comment we took it

• Correct minor typographical issues:

o “This results as ignorance of Tb contact screening.” → “This results in neglect of TB contact screening”

Response: we have great for such type of insightful review, we took it

o “house hold” → household

o “man power” → manpower

o “Up dated” → updated

Response: we corrected these typographical issues:

• Format improvements:

o Bold subtheme heading: “Lack of commitment” under Theme 2.

o Bold full theme headings:

Response: we have made correction on it

Theme 3: Patient / Index Case and Contact–Related Barriers

Theme 4: Social, Cultural, and Economic–Related Barriers to Contact Screening

Response: we have great appreciation for this valuable comment; we have made correction based on the recommended suggestions.

---

## [Decision Letter · Decision Letter 1]

26 Nov 2025

Dear Dr. geremew,

Thank you for submitting your manuscript to PLOS ONE. After careful consideration, we feel that it has merit but does not fully meet PLOS ONE’s publication criteria as it currently stands. Therefore, we invite you to submit a revised version of the manuscript that addresses the points raised during the review process.

Please submit your revised manuscript by Jan 10 2026 11:59PM. If you will need significantly more time to complete your revisions, please reply to this message or contact the journal office at plosone@plos.org . A rebuttal letter that responds to each point raised by the academic editor and reviewer(s). You should upload this letter as a separate file labeled 'Response to Reviewers'.A marked-up copy of your manuscript that highlights changes made to the original version. You should upload this as a separate file labeled 'Revised Manuscript with Track Changes'.An unmarked version of your revised paper without tracked changes. You should upload this as a separate file labeled 'Manuscript'.

We look forward to receiving your revised manuscript.

Kind regards,

Frederick Quinn

Academic Editor

PLOS ONE

Journal Requirements:

Reviewers' comments:

Reviewer's Responses to Questions

**Comments to the Author**

Reviewer #1: (No Response)

2. Is the manuscript technically sound, and do the data support the conclusions?

Reviewer #1: Yes

3. Has the statistical analysis been performed appropriately and rigorously?

Reviewer #1: N/A

4. Have the authors made all data underlying the findings in their manuscript fully available?

Reviewer #1: Yes

5. Is the manuscript presented in an intelligible fashion and written in standard English?

Reviewer #1: Yes

Reviewer #1: 1. The order of "Categories" in Table 2 should align with the order of terms (e.g. . Lack of Human power, Lack of adequate budget, Lack of training ...) are discussed in the text.

2. Check typo carefully. For example "Land scape", "Tb officer".

3. This sentence "To ensure social representativeness,

a range of participants were involved, including the TB Woreda-level coordinator, focal persons,

contact cases, and index cases" appeared twice.

**Do you want your identity to be public for this peer review?** For information about this choice, including consent withdrawal, please see our Privacy Policy

Reviewer #1: No

---

## [Author Response · Author response to Decision Letter 2]

27 Nov 2025

We would like to appreciate your effort to made correction for this paper and here are my responses to the given comments

Reviewer: 1. The order of "Categories" in Table 2 should align with the order of terms (e.g. . Lack of Human power, Lack of adequate budget, Lack of training ...) are discussed in the text.

Response: we would like to say thank you and we made previously mistake on it. We saw these in detail and made correction. The orders and consistent use of words/ phrases in table 2 and discussed was made. Thank you again!!

2. Check typo carefully. For example "Land scape", "Tb officer"

Response : thank you for your insightful view, suggestion, comment and your time to see the typo problems in detailed; we made corrections of these and others in the manuscript.

3. This sentence "To ensure social representativeness,

a range of participants were involved, including the TB Woreda-level coordinator, focal persons,

contact cases, and index cases" appeared twice.

Response: yes there was problem of repeating this sentences twice, we removed the repetition and made correction. Thank you very much!!

---

## [Decision Letter · Decision Letter 2]

15 Dec 2025

Exploring barriers of household contact screening of index case contacts of pulmonary tuberculosis cases in Sekela district, Amhara region, Ethiopia: 2023; descriptive qualitative study

PONE-D-25-44938R2

Dear Dr. Geremew,

We’re pleased to inform you that your manuscript has been judged scientifically suitable for publication and will be formally accepted for publication once it meets all outstanding technical requirements.

Kind regards,

Frederick Quinn

Academic Editor

PLOS One

Additional Editor Comments (optional):

Reviewers' comments:

Reviewer's Responses to Questions

**Comments to the Author**

Reviewer #1: All comments have been addressed

2. Is the manuscript technically sound, and do the data support the conclusions?

Reviewer #1: Yes

3. Has the statistical analysis been performed appropriately and rigorously?

Reviewer #1: N/A

4. Have the authors made all data underlying the findings in their manuscript fully available?

Reviewer #1: Yes

5. Is the manuscript presented in an intelligible fashion and written in standard English?

Reviewer #1: Yes

Reviewer #1: (No Response)

**Do you want your identity to be public for this peer review?** For information about this choice, including consent withdrawal, please see our Privacy Policy

Reviewer #1: No

---

## [Editor Report · Acceptance letter]

PONE-D-25-44938R2

PLOS One

Dear Dr. Geremew,

I'm pleased to inform you that your manuscript has been deemed suitable for publication in PLOS One. Congratulations! Your manuscript is now being handed over to our production team.

Kind regards,

on behalf of

Dr. Frederick Quinn

Academic Editor

PLOS One